# LlamaSeg: Image Segmentation via Autoregressive Mask Generation

## Abstract

We present **LlamaSeg**, a visual autoregressive framework that unifies multiple image segmentation tasks via natural language instructions. By reformulating segmentation as visual generation, LlamaSeg encodes masks as visual tokens and uses a LLaMA-style Transformer for direct next-token prediction, naturally fitting segmentation into autoregressive architectures. To support large-scale training, we introduce a data annotation pipeline and construct the **SA-OVRS** dataset, which contains **2M** segmentation masks annotated with over **5,800** open-vocabulary labels or diverse textual descriptions, spanning diverse real-world scenarios. This enables our model to localize objects in images based on text prompts and to generate fine-grained masks. We further introduce the composite metric average Hausdorff Distance ($d_{\mathrm{AHD}}$) to evaluate mask contour fidelity for generative models better. Experiments show that LlamaSeg consistently outperforms existing generative approaches on multiple segmentation benchmarks and delivers finer, more accurate segmentation masks.

## 1 Introduction

Recent advances in multimodal large language models (MLLMs) have demonstrated the potential of unifying diverse vision-language tasks under an autoregressive framework (Alayrac et al., 2022; Dai et al., 2023; Xu et al., 2024; Zhang et al., 2024). However, extending this paradigm to dense segmentation remains challenging, as it demands precise pixel-level generation (Wang et al., 2023; Wu et al., 2024). As illustrated in Figure 1, existing autoregressive segmentation approaches can be broadly categorized into three paradigms. As illustrated in Figure 1, existing autoregressive segmentation approaches can be broadly grouped into three paradigms. Embedding-based representations (Lai et al., 2024; Xia et al., 2024) predict a latent mask token such as `<SEG>` and then delegate pixel-level mask generation to a specialist model like SAM (Kirillov et al., 2023). Although effective, this reliance on external segmentation experts introduces task-specific components and undermines the goal of a unified end-to-end framework. Coordinate-driven modeling (Wang et al., 2023; Pramanick et al., 2024) encodes each segmentation mask as a sequence of polygon vertices. While naturally aligned with next-token prediction, it requires long coordinate sequences to capture intricate boundaries, reducing efficiency and scalability for dense semantic segmentation. Text-conditioned generation (Wang et al., 2024a; Lan et al., 2024) formulates segmentation as a text generation task by emitting category tokens or free-form descriptions that must later be converted into pixel masks, which inevitably limits spatial precision. These limitations highlight that current autoregressive methods remain inadequate for fine-grained, end-to-end segmentation within a single unified framework.

However, the progress of autoregressive image generation (Esser et al., 2021; Razavi et al., 2019; Sun et al., 2024) shows that appropriate representations and objectives enable these frameworks to model complex visuals. These methods, which typically condition image synthesis on category labels, text prompts, or visual cues, highlight the expressive power of token-based modeling. Their potential remains underexplored in structured prediction tasks with deterministic ground truth, such as segmentation. **Building on these insights, we aim to bridge this gap by revisiting autoregressive modeling for segmentation and devising a representation that predicts high-fidelity masks as discrete visual tokens within a unified next-token prediction framework.**

In this paper, we propose **LlamaSeg**, which reformulates image segmentation as a visual generation task, as shown in panel (d) of Figure 1. We treat segmentation masks as image-like structures, which

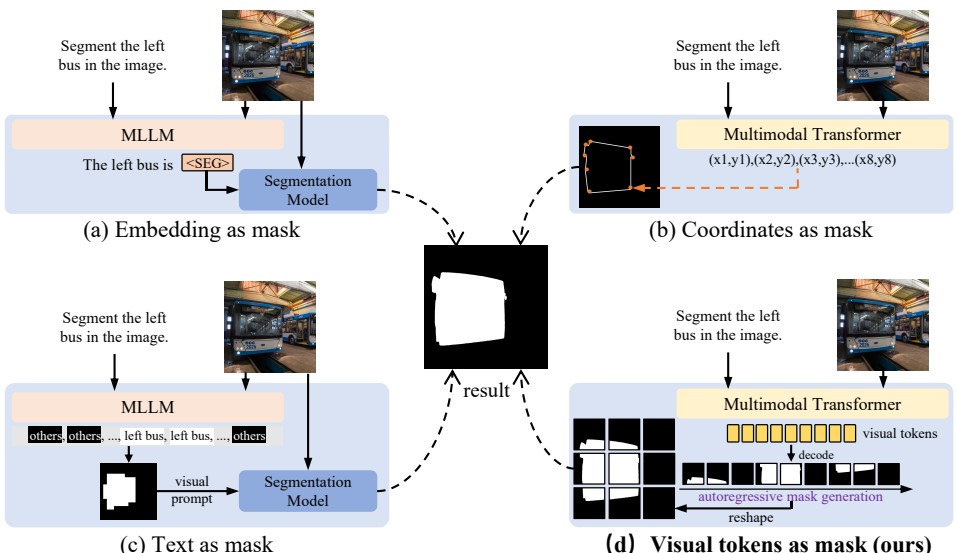

Figure 1: Comparison of autoregressive segmentation paradigms. (a) Embedding as mask. (b) Coordinates as mask. (c) Text as mask. (d) Visual tokens as mask (ours), which directly autoregresses visual tokens to capture fine-grained structures and yields precise pixel-level masks.

are discretized into token sequences via VQGAN (Esser et al., 2021). A LLaMA-based autoregressive model (Touvron et al., 2023a;b) then generates mask tokens conditioned on the encoded visual inputs, maintaining the standard next-token prediction paradigm, thereby facilitating the integration of segmentation tasks into general autoregressive frameworks. Our framework is scalable across different model sizes and image resolutions, and supports training from scratch as well as fine-tuning on pretrained MLLMs. To address the lack of suitable training data and evaluation tools for this paradigm, we introduce a large-scale dataset, SA-OVRS, containing 2 million segmentation instances annotated with open-vocabulary labels or rich textual descriptions derived from SA-1B (Kirillov et al., 2023), and propose the average Hausdorff Distance ($d_{\mathrm{AHD}}$) under multi-level IoU thresholds as a new metric to assess contour fidelity in generated masks. Experiments on multiple segmentation benchmarks demonstrate that our model produces high-quality, fine-grained masks and outperforms existing visual generative approaches. Our main contributions are as follows:

1. We propose **LlamaSeg**, a visual autoregressive segmentation framework that shares the core next-token generation paradigm of Large Language Models (LLMs), allowing segmentation to be natively unified within the LLMs architecture while effectively modeling visual context and producing high-quality, fine-grained masks.

2. We develop a data annotation pipeline and construct the **SA-OVRS** dataset comprising 2M masks with over 5,800 open-vocabulary or richly descriptive labels. The dataset and the associated annotation pipeline are released as reusable tools to advance future segmentation research and to facilitate the creation of new high-quality datasets.

3. We introduce a composite evaluation metric $d_{\mathrm{AHD}}$ to assess mask-contour fidelity and demonstrate that LlamaSeg surpasses most existing visual generative models on diverse semantic and referring segmentation benchmarks, delivering more precise pixel-level masks.

## 2 RELATED WORK

### 2.1 MULTIMODAL LARGE LANGUAGE MODELS

The research on MLLMs is closely related to cross-modal representation learning. With the rapid advancement of LLMs (Vaswani et al., 2017; Radford et al., 2018; Brown et al., 2020; Chowdhery et al., 2023), recent efforts have increasingly focused on aligning visual representations with the language embedding space. BLIP-2 (Li et al., 2023) and InstructBLIP (Dai et al., 2023) incorporate lightweight Querying Transformer modules to inject visual context. The LLaVA series (Liu et al., 2023b; 2024a; Li et al., 2024) employs simple linear projection layers to map image features. The Qwen-VL family (Bai et al., 2023; Wang et al., 2024b) leverages cross-attention mechanisms with

learnable queries for tighter vision-language integration. These approaches collectively enable large language models to interpret and reason over visual content effectively. Beyond image-level understanding, recent studies have begun to extend vision-language models toward dense prediction tasks such as image segmentation, which require fine-grained spatial reasoning and localized visual grounding.

### 2.2 Image Segmentation

**Transformer-based segmentation method.** Segmentation models based on the Transformer (Vaswani et al., 2017) architecture typically include MaskFormer (Cheng et al., 2021), SegFormer (Xie et al., 2021), and SAM (Kirillov et al., 2023). Since language models such as BERT (Devlin et al., 2019) also adopt Transformer architectures, this shared foundation facilitates cross-modal information fusion. Several approaches (Ding et al., 2021; Wang et al., 2022; Liu et al., 2023a) have explored referring image segmentation built upon this architecture. Recent studies have increasingly focused on MLLMs. For example, LISA (Lai et al., 2024) integrates SAM and LLaVA (Liu et al., 2023b), leveraging token embeddings to guide the segmentation model. Subsequent methods (Ren et al., 2024; Xia et al., 2024) build upon this framework with further enhancements. Despite their strong performance, these approaches typically adopt a pipeline architecture, which increases model complexity and training difficulty.

**Generative segmentation method.** Generative methods recast segmentation as a mask-generation problem conditioned on the input image, providing a streamlined alternative to traditional pipeline-based approaches. GSS (Chen et al., 2023), for example, links the mask posterior distribution with the latent prior of the input image for semantic segmentation. Despite adopting the generative paradigm, these approaches remain confined to vision-only tasks. Unified-IO (Lu et al., 2022) and Unified-IO2 (Lu et al., 2024) extend to multiple modalities and include segmentation, but neither explores it deeply. Unified-IO struggles with complex language understanding, while Unified-IO2 employs a two-stage procedure dependent on bounding boxes. Such reliance on task-specific modules or intermediate representations hinders seamless end-to-end segmentation and limits generalization.

### 2.3 Autoregressive Image Generation

Autoregressive modeling enables end-to-end image synthesis pipelines, offering an appealing alternative to traditional modular designs. Recent models (Tian et al., 2024; Team) discretize images into 1D sequences with image tokenizers for next-token prediction. Subsequent work (Yu et al., 2021; Sun et al., 2024) refines tokenizer design and benchmarks against continuous models such as VAE (Kingma et al., 2013), showing competitive generation quality. Despite these advances, applying autoregressive models to dense prediction tasks like segmentation remains largely unexplored.

## 3 LlamaSeg

### 3.1 Overview

The overall framework of LlamaSeg is illustrated in Fig. 2. For the LLaMA-based framework, we employ image and text encoders to extract input features, which are projected to align with the embedding dimension of LLaMA (Touvron et al., 2023a;b) and concatenated with learnable separator tokens to form the input sequence. The LLaMA model then autoregressively generates mask tokens as output. We adopt VQGAN (Esser et al., 2021) as the mask tokenizer. During training, the mask tokenizer encodes the ground truth mask into code indices, which serve as the training targets for the LLaMA model. During inference, the mask tokenizer receives the mask tokens generated by the LLaMA model, retrieves the corresponding codes from the codebook, and decodes them to produce the segmentation mask. For the MLLM-based framework utilizing a pretrained MLLM, the text encoder is omitted, while the remaining procedure remains unchanged.

### 3.2 Mask Tokenizer

We use an image tokenizer to transform segmentation masks into discrete tokens. Specifically, we use a VQGAN from LlamaGen (Sun et al., 2024) with a downsample rate of 16 and a codebook $Z \in \mathbb{R}^{K \times d_{vq}}$ containing $K$ vectors, where $d_{vq}$ is the vector dimension. We consider the segmentation mask as a special RGB image composed only of black and white pixels. During training, the encoder $\mathcal{E}$ transforms the normalized mask $M_I \in \mathbb{R}^{H \times W \times 3}$ into features $f_m \in \mathbb{R}^{h \times w \times d_{vq}}$, and the quantizer $\mathcal{Q}$ converts features into discrete tokens $q \in [K]^{h \times w}$. The quantizer matches each vector in $f_m$ with

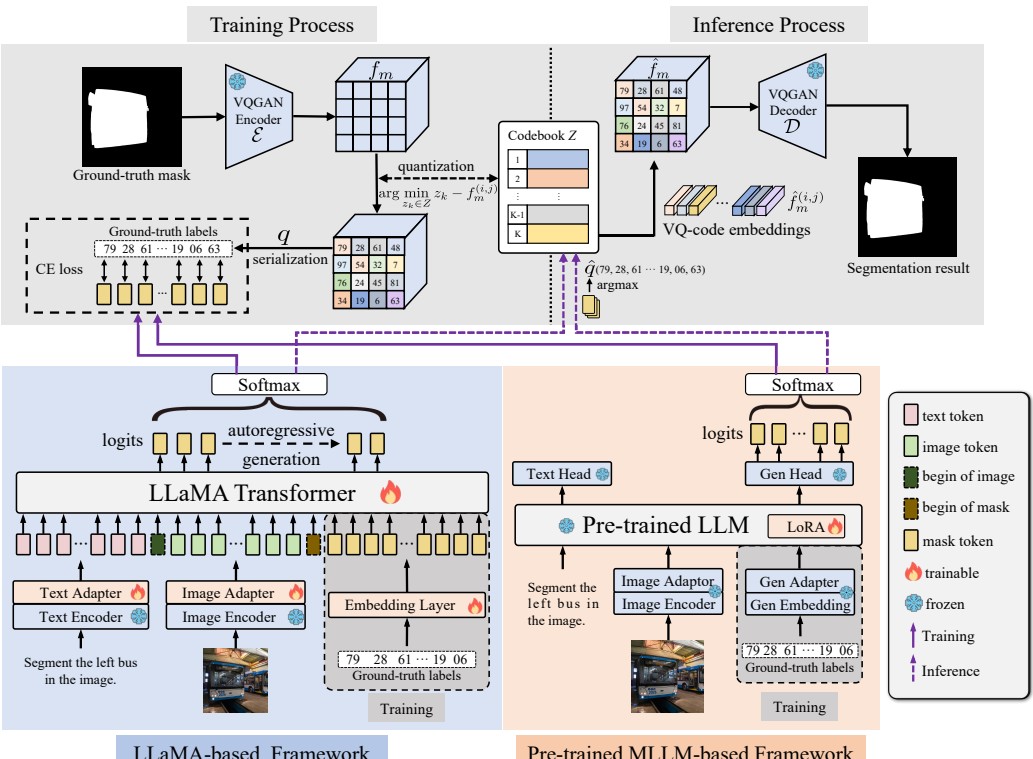

Figure 2: Overall framework of **LlamaSeg**. The model consists of a VQGAN-based mask tokenizer and a LLaMA-style autoregressive generator. During training, ground-truth masks are quantized into discrete codes to supervise token prediction; during inference, predicted tokens are decoded to produce segmentation masks. We support training either from scratch or by adapting a pre-trained MLLM with lightweight modules.

the closest vector in $Z$ in terms of Euclidean distance and finds the corresponding code index:

$$q^{(i,j)} = \arg\min_{z_k \in Z} ||z_k - f_m^{(i,j)}|| \in [K], \tag{1}$$

where $z_k$ is the vector within $Z$. Flattening the code indices to form a 1D sequence can serve as supervised training data for an autoregressive model. At inference time, for a 1D token output by an autoregressive model, the token IDs are equivalent to the code indices in the mask tokenizer. By "looking up in table", vectors $\hat{z}_k \in \mathbb{R}^{d_{vq}}$ can be found in the codebook according to $\hat{q} \in [K]^{h \times w}$, which are rearranged into a 2D shape from token IDs and form the feature map $\hat{f}_m \in \mathbb{R}^{h \times w \times d_{vq}}$. The normalized mask image $\hat{M}_I \in [-1, 1] \subset \mathbb{R}^{H \times W \times 3}$ is reconstructed by the decoder $\mathcal{D}$ using $\hat{f}_m$. Average $\hat{M}_I$ across the channel dimension and then apply a binarization step to produce the final segmentation result $\hat{M}_{bin}^{(i,j)}$:

$$\hat{M} = \frac{1}{3} \sum_{i=0}^{2} (\hat{M}_I[:,:,i]), \qquad \hat{M}_{bin}^{(i,j)} = \begin{cases} 1, & \hat{M}^{(i,j)} \geq 0 \\ 0, & \text{otherwise} \end{cases}. \tag{2}$$

### 3.3 IMAGE AND TEXT FEATURE EXTRACTION

We construct textual inputs by combining object labels or descriptions with 10 predefined templates, such as "Produce a segmentation mask for the {*object name*}." For the autoregressive model trained from scratch, we employ a frozen SigLIP2 (Tschannen et al., 2025) with a patch size of 16 to encode both images and text. The vision encoder's patch size matches the mask tokenizer's downsampling rate, ensuring that input tokens correspond to the same pixel regions as the segmentation mask for precise token-level alignment. Furthermore, trainable adaptors are used to project the image and text features into the embedding dimension of the autoregressive model. Each adaptor has two linear layers and a GELU (Hendrycks & Gimpel, 2016) activation function. We consider visual encoders with input resolutions of 256 and 384, corresponding to 256 and 576 visual tokens, respectively.

### 3.4 AUTOREGRESSIVE MODEL

**LLaMA Structure.** The autoregressive model is based on LLaMA (Touvron et al., 2023a;b) model Each Transformer layer employs 1D RoPE, treating both images and masks as 1D sequences. We utilize two model variants of different sizes. The base model comprises 770 million parameters, 16 layers, a hidden size of 1920, and 20 attention heads. The large model contains 1.5 billion parameters, 22 layers, a hidden size of 2304, and 32 attention heads. During training, input image and text features are concatenated along the channel dimension. To distinguish between text, image, and mask tokens, we introduce learnable separators: `<BOI>` (*Begin-Of-Image*) and `<BOM>` (*Begin-Of-Mask*). The input sequence during training is structured as follows:

```
[text tokens]<BOI>[image tokens]<BOM>[mask tokens].
```

The model is trained using a cross-entropy loss, computed solely on the mask tokens. At inference time, only `<BOM>` and the preceding tokens are fed into the model, allowing it to generate mask tokens autoregressively.

**Utilization of pre-trained MLLM.** We adopt Janus Pro (Chen et al., 2025), an MLLM for multi-modal understanding and generation with separate language and image pathways. For multimodal understanding, images are encoded by the SigLIP (Zhai et al., 2023) vision encoder. For image generation, however, visual inputs are synthesized via an image tokenizer (Esser et al., 2021). The model operates at an image resolution of 384. To enable visual generation, we utilize components responsible for producing mask tokens, denoted as "Gen Embedding", "Gen Adapter", and "Gen Head" in Fig. 2. These components are kept frozen to retain the generative capabilities acquired through large-scale data training. During training, the sequence input to the model is:

**<|User|>:** [text tokens]`<image>` **<|Assistant|>:** `<begin_of_mask>`[mask tokens]

where `<|User|>` and `<|Assistant|>` denote the dialogue roles, `<image>` represents the input image, and `<begin_of_mask>` indicates the beginning of the mask token sequence. During inference, only `<begin_of_mask>` and its preceding context are provided as input.

## 4 SA-OVRS DATASET CONSTRUCTION

Existing methods (Lai et al., 2024; Xia et al., 2024) leverage semantic and referring segmentation datasets for language-guided segmentation. Semantic datasets such as COCO-Stuff (Caesar et al., 2018) treat category names as text, but hierarchical labels (e.g., "vegetables" vs. "broccoli") are split into independent classes, leading to semantic inconsistencies and limiting open-vocabulary learning. In addition, existing datasets are too small to support the dense cross-modal alignment required by autoregressive models. Figure 3 illustrates the two-stage annotation pipeline used to construct SA-OVRS. In the first stage, we generate candidate open-vocabulary labels for SA-1B (Kirillov et al., 2023) images and match them with their corresponding masks through quality filtering, while the second stage generates natural-language descriptions for each object instance, encompassing both referring and reasoning expressions. More details are presented in the following subsection.

### 4.1 LABEL GENERATION AND MASK MATCHING

For each SA-1B image, we first obtain candidate open-vocabulary labels using Qwen2-VL-72B with a tailored prompt, limiting outputs to at most ten labels per image. GroundingDINO (Liu et al., 2024b) then detects bounding boxes for each label. To ensure high-quality matches, we apply a three-step filtering procedure: **(i)** Remove labels associated with excessive boxes (more than four). **(ii)** Eliminate nested detections where a smaller box is almost entirely enclosed in a larger one (IoU > 0.97). **(iii)** Keep only matches with sufficient confidence (score > 0.3) and strong mask overlap (IoU > 0.85 for single-box labels, 0.9 for multi-box labels). Masks linked to the same label are merged to produce semantic segmentation samples.

### 4.2 TEXTUAL DATA GENERATION AND VERIFICATION

To produce unique natural-language descriptions for each instance, we highlight every matched mask with a green contour and prompt Qwen2-VL-72B to generate a referring expression that distinguishes it from all other objects in the image. We further perform cross-verification by recoloring the mask

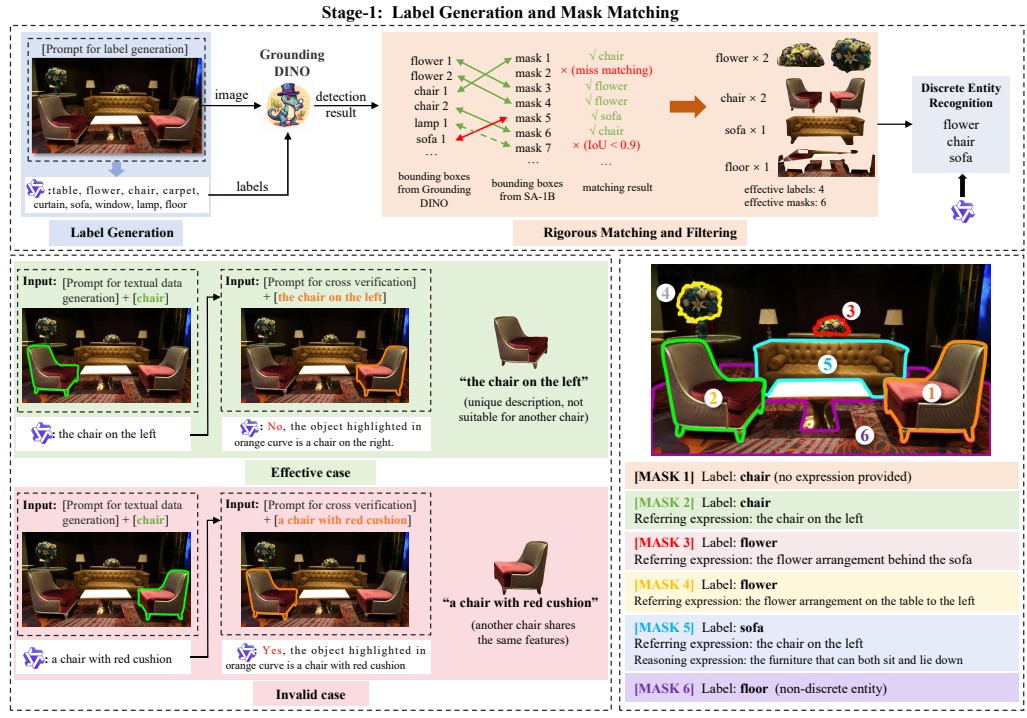

Figure 3: A two-stage annotation pipeline for the construction of the **SA-OVRS** dataset. **Stage 1: Label Generation and Mask Matching** (top) leverages Qwen2-VL to generate candidate open-vocabulary labels, which are then aligned with SA-1B masks using GroundingDINO through strict matching and filtering procedures. **Stage 2: Textual Data Generation and Verification** (bottom left) produces high-quality referring expressions and reasoning descriptions.

contour and prompting the model to confirm that the expression is unique and unambiguous. This ensures high-quality referring segmentation annotations.

To further enhance data diversity, we introduce reasoning segmentation. Unlike referring expressions that directly identify objects, reasoning expressions omit object names and instead describe them by function or commonsense attributes (e.g., "the furniture that can both sit and lie down" → sofa). This design encourages models to leverage semantic reasoning beyond surface lexical cues. This process differs in prompt design and omits the verification step.

In total, SA-OVRS contains **1.93M validated instance masks, 1.15M semantic samples, and 850K textual expressions (800K referring, 50K reasoning)** spanning over 5,800 open-vocabulary labels.

## 5 EXPERIMENTS

### 5.1 EXPERIMENT SETTINGS

**Training Data.** For semantic segmentation, we select ADE20K (Zhou et al., 2019) and COCO-Stuff (Caesar et al., 2018) dataset. For referring segmentation, we choose the refCOCO (Yu et al., 2016) series and refCLEF (Kazemzadeh et al., 2014) datasets. Additionally, we adopt the semantic and referring segmentation data from SA-OVRS.

**Tasks and Evaluation Metrics.** For closed-set and open-vocabulary semantic segmentation, we employed mean IoU (mIoU). For referring segmentation, we follow prior works and use the cumulative intersection over cumulative union (cIoU). To evaluate the contour accuracy of masks produced by visual generative models, we propose the *average Hausdorff Distance ($d_{AHD}$)* metric under multi-level IoU thresholds. Given the boundary point sets of a predicted mask and GT mask, $d_{AHD}$ is defined as:

$$d_{AHD}(X, Y) = \frac{1}{2}\left(\frac{1}{X}\sum_{x \in X}\min_{y \in Y} dist(x, y) + \frac{1}{Y}\sum_{y \in Y}\min_{x \in X} dist(x, y)\right), \quad (3)$$

Table 1: Comparisons with visual generative models and MLLMs on closed-set and open-vocabulary semantic segmentation datasets. Unified-IO and Unified-IO2 are evaluated using their open-source weights. "Params" denotes model size and "384 pix." indicates a 384-pixel input resolution. **Bold** marks the best result.

| Type | Model | Params | ADE20K | COCO-Stuff | PC-459 | PC-59 | PAS-20 |
|------|-------|--------|--------|------------|--------|-------|--------|
| MLLM | LaSagnA (Wei et al., 2024) | 7B | 42.0 | 43.9 | 9.8 | 39.6 | 61.8 |
| Visual generative model | Unified-IO-Large | 0.77B | 39.9 | 50.7 | - | 62.8 | 62.7 |
| | Unified-IO-XL | 2.9B | 45.2 | 55.2 | - | **64.0** | 74.9 |
| | Unified-IO2-Large | 1.1B | 35.6 | 48.8 | - | 55.8 | 71.5 |
| | Unified-IO2-XL | 3.2B | 39.4 | 49.9 | - | 56.9 | 71.5 |
| | Unified-IO2-XXL | 6.6B | 51.9 | 55.1 | - | - | - |
| Visual generative model (ours) | **LlamaSeg-B** | 0.77B | 50.5 | 54.5 | 28.5 | 61.7 | 74.5 |
| | **LlamaSeg-1B**(MLLM) | 1B | 45.1 | 50.1 | **35.6** | 58.2 | 71.1 |
| | **LlamaSeg-L** | 1.5B | **52.0** | **55.7** | 35.5 | 62.5 | **75.1** |
| | **LlamaSeg-L**(384 pix.) | 1.5B | 55.9 (+3.9) | 58.1 (+2.4) | 36.7 (+1.2) | 63.8 (+1.3) | 77.1 (+2.0) |

Table 2: Comparison results on referring segmentation datasets. We evaluated the performance of Unified-IO and Unified-IO2 on a subset of datasets. Green indicates the best result among discriminative segmentation models, while orange indicates the best result among visual generative models.

| Type | Model | Params | refCOCO | | | refCOCO+ | | | refCOCOg | |
|------|-------|--------|---------|---|---|----------|---|---|----------|---|
| | | | val | testA | testB | val | testA | testB | val | test |
| Discriminative segmentation model | LAVT (Yang et al., 2022) | - | 72.7 | 75.8 | 68.8 | 62.1 | 68.4 | 55.1 | 61.2 | 62.1 |
| | ReLA (Liu et al., 2023a) | - | 73.8 | 76.5 | 70.2 | 66.0 | 71.0 | 57.7 | 65.0 | 66.0 |
| Visual generative model | Unified-IO-Large | 0.77B | 35.7 | 39.2 | - | 34.7 | 39.4 | - | 42.2 | 42.3 |
| | Unified-IO-XL | 2.9B | 42.4 | 49.5 | - | 39.4 | 42.7 | - | 50.3 | 50.3 |
| | Unified-IO2-Large | 1.1B | 40.3 | 47.2 | - | 33.0 | 39.1 | - | 43.2 | 44.1 |
| | Unified-IO2-XL | 3.2B | 50.7 | 55.7 | - | 39.4 | 47.4 | - | 52.6 | 54.5 |
| | Unified-IO2-XXL | 6.6B | 54.8 | - | - | 44.8 | - | - | - | - |
| Visual generative model (ours) | **LlamaSeg-B** | 0.77B | 50.9 | 56.1 | 38.9 | 38.9 | 44.5 | 32.9 | 51.0 | 50.5 |
| | **LlamaSeg-1B**(MLLM) | 1B | 52.3 | 57.8 | 47.1 | 44.6 | 51.6 | 36.5 | 49.0 | 49.7 |
| | **LlamaSeg-L** | 1.5B | 56.5 | 60.5 | 52.6 | 41.6 | 46.2 | 36.2 | 51.0 | 50.4 |

where $dist(x, y)$ denotes the Euclidean distance between x and y. This metric captures bidirectional distances between the two point sets, providing an aggregate measure of global boundary alignment, where a **lower score** indicates better performance. For each threshold in [0.5, 0.6, 0.7, 0.8, 0.9], we collect predictions with IoU above it, compute $d_{AHD}$, and report the mean average Hausdorff distance ($mAHD$)

**Pre-training.** We adopt a two-stage training protocol consisting of pre-training followed by fine-tuning. During pre-training, the model is trained on the SA-OVRS dataset and the referring segmentation datasets. We use the AdamW (Loshchilov & Hutter, 2017) optimizer with a learning rate of $2 \times 10^{-4}$, $\beta_1 = 0.9$, $\beta_2 = 0.95$, weight decay 0.05, and run for 4 epochs.

**Fine-tuning.** The semantic and referring segmentation tasks are trained separately on their respective datasets. We use AdamW with a learning rate of $1 \times 10^{-4}$, $\beta_1 = 0.9$, $\beta_2 = 0.99$, zero weight decay, and a WarmupCosineDecay scheduler with 1% linear warmup. We fine-tune for 10 epochs on the semantic segmentation datasets and 20 epochs on the referring segmentation datasets. All training is carried out on 8 NVIDIA GPUs (A800 or H20) under PyTorch's distributed data parallel framework.

**Model Variants.** For the LLaMA-based framework, mask tokens are generated via greedy search. When leveraging an MLLM-based framework, we apply LoRA (Hu et al., 2022) for efficient fine-tuning and remove the unconditional generation component while keeping other configurations unchanged. We develop two models within the LLaMA-based framework (from scratch), namely **LlamaSeg-B (base)** and **LlamaSeg-L (large)**, and further construct **LlamaSeg-1B**(MLLM) within the Pre-trained MLLM-based Segmentation Framework (with LoRA/Adapter).

## 5.2 MAIN RESULTS

**Semantic Segmentation Result Analysis.** Table 1 reports closed-set and open-vocabulary semantic segmentation results. LlamaSeg consistently outperforms existing visual generative models across all benchmarks. LlamaSeg-L attains 52.0/55.7 mIoU on ADE20K/COCO-Stuff, surpassing Unified-IO-XL and Unified-IO2-XXL by large margins with fewer parameters. Using higher-resolution mask tokens, LlamaSeg-L (384 pix.) further improves to 55.9 (+3.9) and 58.1 (+2.4), with consistent gains on PC-459/PC-59/PAS-20 (e.g., +2.0 on PAS-20). These results show that our visual-token

Table 3: Comparison of the $mAHD$ metric after normalization to a 256-pixel resolution. Lower values correspond to a closer match between the predicted mask contours and the ground truth. Bold entries highlight the best results.

| Model | ADE20K | | | | | refCOCO | | | | |
|---|---|---|---|---|---|---|---|---|---|---|
| | IoU-0.5 | IoU-0.6 | IoU-0.7 | IoU-0.8 | IoU-0.9 | IoU-0.5 | IoU-0.6 | IoU-0.7 | IoU-0.8 | IoU-0.9 |
| Unified-IO-Base | 76.41 | 74.84 | 72.36 | 68.72 | 62.39 | 85.14 | 83.22 | 81.26 | 80.02 | 79.06 |
| Unified-IO-Large | 76.35 | 75.31 | 72.21 | 67.73 | 62.62 | 83.08 | 81.02 | 79.81 | 78.38 | 75.61 |
| Unified-IO-XL | 75.88 | 75.16 | 72.35 | 69.35 | 65.13 | 72.58 | 71.69 | 71.16 | 69.99 | 85.65 |
| Unified-IO2-Large | 78.45 | 76.96 | 75.02 | 72.62 | 67.37 | 83.32 | 82.57 | 81.68 | 80.59 | 79.14 |
| Unified-IO2-XL | 80.15 | 78.51 | 76.48 | 74.21 | 67.82 | 82.05 | 81.42 | 80.91 | 79.95 | 78.18 |
| Unified-IO2-XXL | 79.72 | 78.02 | 75.76 | 71.81 | 65.96 | 80.98 | 80.10 | 79.45 | 78.24 | 75.33 |
| **LlamaSeg-B** | 26.61 | 25.91 | 25.61 | 25.98 | 29.28 | 14.40 | 13.63 | 12.69 | 11.36 | 9.94 |
| **LlamaSeg-1B**(MLLM) | 59.24 | 58.91 | 59.20 | 60.77 | 69.38 | 45.47 | 42.41 | 39.29 | 37.80 | 37.28 |
| **LlamaSeg-L** | **25.54** | **24.73** | **24.28** | **24.40** | **26.51** | **10.45** | **9.68** | **8.73** | **7.42** | **5.27** |

formulation scales well with model size and resolution, enabling fine-grained masks and better boundary preservation. Compared with the MLLM-based baseline (LaSagnA) and the Unified-IO series, LlamaSeg achieves higher accuracy while remaining fully autoregressive and end-to-end.

**Referring Segmentation Result Analysis.** As shown in Table 2, our model outperforms existing visual generative models across multiple referring segmentation datasets under the same parameter constraints. On the RefCOCO dataset, our LlamaSeg-Large achieves a score 1.7 points higher than the best result of Unified-IO2, demonstrating superior language understanding capabilities. In contrast, a noticeable gap remains compared to specialized discriminative segmentation models.

**Boundary Accuracy Analysis.** Although our model achieves slightly lower overall scores on referring segmentation tasks than other strong baselines, it delivers notably higher boundary precision. We evaluate $mAHD$ under multiple IoU thresholds on ADE20K and RefCOCO (Table 3). While semantic segmentation data—aimed at pixel-level discrimination—often exhibits complex contours, referring segmentation masks are comparatively simpler. Our approach consistently yields lower $mAHD$ than previous visual generative models, indicating more accurate alignment with ground-truth contours and the ability to capture fine object boundaries with high fidelity. Figure 4 further illustrates these improvements in mask quality and segmentation accuracy.

**2D Spatial Relationship.** To examine whether the model can learn 2D spatial relationships from 1D tokens, we select an example in resolution of 256 and export the attention map in the last self-attention layer of the Transformer model, as shown in Fig. 5. Under the causal attention mechanism, the attention map exhibits a distinct band parallel to the main diagonal, revealing that mask tokens preferentially attend to tokens located in preceding rows but within the same column in the original 2D layout. Moreover, the presence of regular dark vertical stripes indicates that, despite

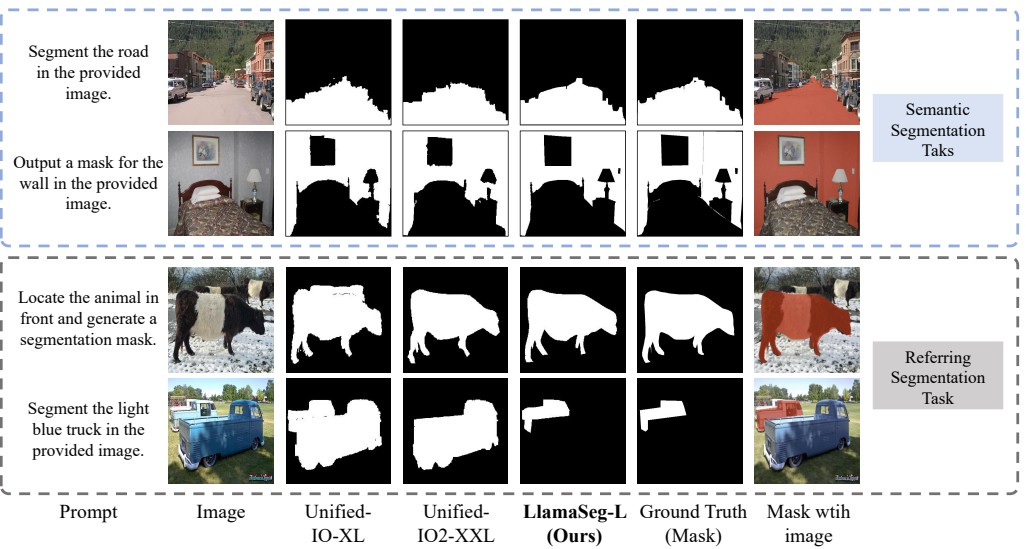

Figure 4: Visual comparison between our model and other visual generative models.

the linear adjacency of row-end and next-row tokens in the 1D sequence, the model consistently suppresses attention from the leading tokens of each row to the trailing tokens of the previous row.

These observations demonstrate that our method does not merely exploit 1D sequential proximity but internally reconstructs and leverages authentic 2D spatial topology.

### 5.3 ABLATION STUDY

**Mask Tokenizer Analysis.** We further fine-tune the pre-trained VQGAN on segmentation masks with a batch size of 128. The overall IoU and $mAHD$ results (without IoU threshold constraints) for mask reconstruction are reported in Table 4. While incorporating additional training data slightly improves reconstruction quality, the overall performance remains below that of the original pre-trained model, suggesting that extended fine-tuning leads to a degradation of the learned feature representations.

**Decoding Strategy Analysis.** We conduct experiments on the RefCOCO validation set using the LlamaSeg-B model to systematically compare several decoding strategies, as reported in Table 5. Given that image segmentation tasks are paired with deterministic ground-truth masks, the model is required to produce a single, optimal prediction rather than diverse outputs. These results demonstrate that greedy decoding is the most effective and reliable strategy for autoregressive image segmentation in this setting.

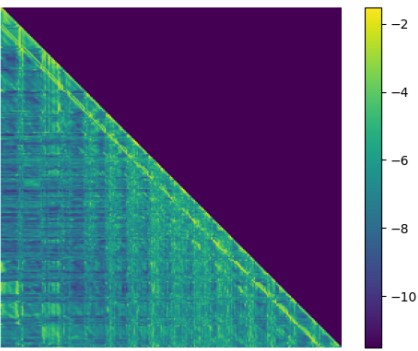

Figure 5: Attention heatmap of mask tokens in the last self-attention layer of the autoregressive model. Scores have been log-transformed to enhance visual contrast.

Table 4: Ablation study on mask tokenizer. **Bold** indicates the best result.

| Train steps | Total IoU↑ | mAHD↓ |
|---|---|---|
| 0 (frozen) | **96.8** | **6.1** |
| 5000 | 93.0 | 12.5 |
| 10000 | 93.5 | 10.6 |

Table 5: Ablation study on decoding strategy. The IoU threshold of mAHD is set to 0.5.

| Decoding strategy | RefCOCO | |
|---|---|---|
| | cIoU↑ | mAHD↓ |
| Greedy search | **50.9** | **14.4** |
| Beam search (B=3) | 47.3 | 15.1 |
| Top-K (K=3) | 49.4 | 15.3 |
| Top-P (P=0.9) | 50.2 | 15.3 |
| Random sample | 49.2 | 15.7 |

Table 6: Ablation study on training data. "Sem." denotes semantic segmentation data and "Ref." denotes referring segmentation data. The IoU threshold of mAHD is set to 0.5.

| Pre-training data | Fine-tuning data | ADE20K | | refCOCO | |
|---|---|---|---|---|---|
| | | mIoU↑ | mAHD↓ | cIoU↑ | mAHD↓ |
| SA-OVRS+Ref. | Sem. | **50.5** | **26.6** | - | - |
| - | Sem. | 48.9 | 28.7 | - | - |
| SA-OVRS+Ref. | Ref. | - | - | **50.9** | **14.4** |
| - | Ref. | - | - | 46.0 | 16.2 |

**SA-OVRS Contribution Analysis.** To assess the contribution of the SA-OVRS dataset to segmentation performance, we train the LlamaSeg-B model with varying combinations of training data and report the results in Table 6. The ablation shows a clear performance drop when pre-training data are omitted, highlighting the importance of large-scale pre-training for capturing rich visual representations. Nevertheless, even without pre-training data, our model still surpasses existing visual generative models of comparable parameter size.

## 6 CONCLUSION

In this paper, we propose **LlamaSeg**, a novel visual autoregressive image segmentation method that integrates segmentation into a standard autoregressive framework. We further introduce a two-stage data annotation pipeline to construct **SA-OVRS**, a large-scale open-vocabulary segmentation dataset containing 2M high-quality samples, which provides valuable resources for training and future research. Extensive experiments show that LlamaSeg not only surpasses existing visual generative models of comparable size across multiple benchmarks but also provides a scalable and unified paradigm that can inspire future multimodal and autoregressive segmentation research.

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
