**Disclosure of LLM usage.** This paper benefited from language editing and phrasing suggestions provided by ChatGPT (OpenAI), which was used solely for grammar correction and clarity improvement. No LLM was used for generating research ideas, experimental design, data analysis, or writing substantive technical content.

## A  DETAILS OF DATA ANNOTATION

### A.1  PROMPTS FOR DATA ANNOTATION

```
Identify up to ten distinct prominent objects in the image on their
shape and appearance prioritized by prominence. Output these nouns in
singular form using the format: word.word.word... Avoid repetition,
extra punctuation, or newlines.
```

Figure 1: Prompt for label generation.

```
**Task**: Generate unique referring expressions for the **{ann["label"]}** highlighted with green edges. Follow
this decision hierarchy:

1. **Dual-cue Fusion Principle**
- Analyze *the most prominent visual distinctiveness* (size, color, state) and *spatial
relationships*(top/bottom/left/right, inside, front) that can uniquely point to the green-edged target
- Combine visual distinctiveness and spatial relationships, unless one of them is sufficient to provide clear
identification

2. **Object Identification Protocol**
- **Must reference the green-edged target** (failure will invalidate output)
- **Understand green edges are temporary markers for your attention guidance** (Never describe the green edges as
object attributes)
- **Do not references textual content on objects** (e.g. text/number on clothing/signs, trademarks or company name)
- **Generate referring expressions using visual distinctiveness and spatial relationships, instead of motions**
- **Same-class objects are highlighted with orange edges.** When multiple same-class objects exist, visual
distinctiveness and spatial relationships must distinguish the green edges target with other orange edges targets

3. **Output Format**
- Strict template:
```
Highlighted in green: [visual-spatial fusion descriptor]
```
- Examples revised:
✓ Highlighted in green: the woman in all white
✓ Highlighted in green: the purple bottle in corner
✓ Highlighted in green: the largest gift box with a blue ribbon
✓ Highlighted in green: the giraffe standing to the left of a tree
```

Figure 2: Prompt for referring expression generation.

```
### System Instructions
You are a visual analysis expert. Carefully inspect the image where a specific object is highlighted with bright
orange edges.
Cross-reference the given textual clues to determine whether the provided word and description accurately and
uniquely correspond to the highlighted object.
Respond with precise logical reasoning.

### Input Context
* **Target word**: `{ann["label"]}`
* **Text description**: `{ref_desc}`
* **Visual cue**: The object is outlined with an orange contour line

### Output Format
- You should use the following output format:
```
Conclusion: [Yes/No]. Reasons: [Analysis]
```
```

Figure 3: Prompt for referring expression verification.

```
**Task**: Generate a concise and unambiguous **reasoning expression** for the **{label}** highlighted with green edges in
the image. Your expression should rely on visual and spatial reasoning, but **must not include the object's category or type
name**.

### Internal Reasoning Protocol (for guidance only, not to be output):
1. **Object Recognition (Internally)**
   – Determine what the object is based on appearance, but do **not** name or mention it.

2. **Visual Feature Identification**
   – Identify features such as color, size, texture, shape, posture, and material that make the object unique.

3. **Spatial Reasoning**
   – Use spatial cues (e.g., near the corner, behind another object, to the left of something) to enhance precision.

4. **Disambiguation**
   – If other similar-looking objects exist, highlight what distinguishes this one using **only visual and spatial cues** – no
   category names.

5. **Expression Construction**
   – Formulate a **natural phrase** that uniquely identifies the object **without using its class name**.

### Constraints:
   – Describe only the green-highlighted object.
   – **Do not mention green edges**.
   – **Do not mention any visible text** (e.g., logos, numbers).
   – **Do not refer to object motion**.
   – **Do not include object category words** (e.g., person, bottle, dog, book, etc.).
   – Rely **only on visual attributes and spatial layout** to identify the object.

### Additional Style Constraints :
   – Do **not** begin the expression with vague placeholders like "the object", "the thing", or "the item".
   – Instead, begin the phrase directly with **visual features or spatial descriptors**, such as:
      – "the one in a red coat near the bench"
      – "the glossy one to the left of the big box"
      – "the smallest one under the table"
```

Figure 4: Prompt for reasoning expression generation.

## A.2  LABEL VISUALIZATION

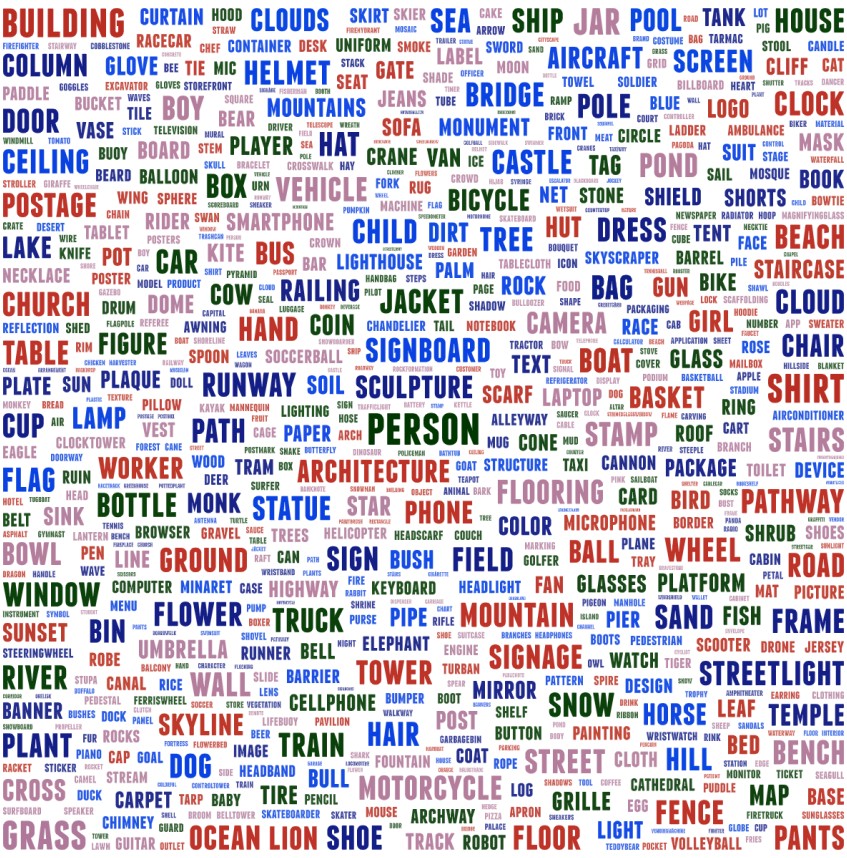

Figure 5: Visualizations of selected labels from the SA-OVRS dataset. The size of each label roughly reflects
its abundance in the dataset.

## A.3 Data Examples

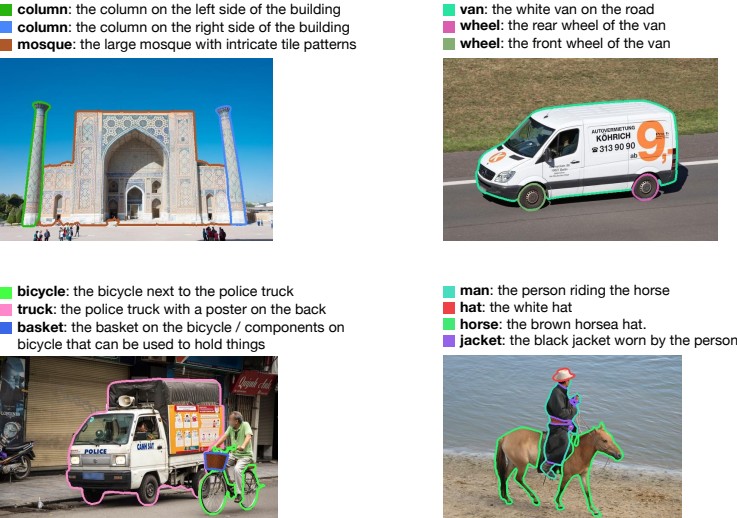

Figure 6: Visualizations of selected labels and textual descriptions from the SA-OVRS dataset.

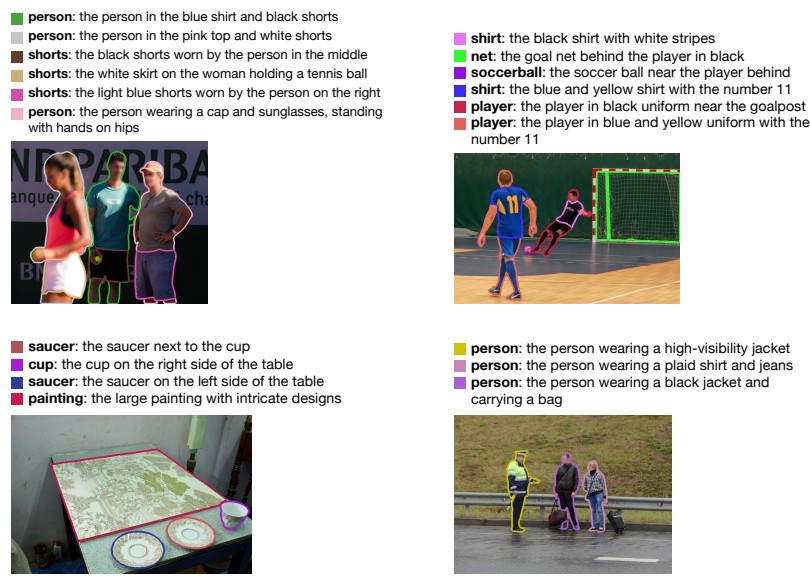

Figure 7: Visualizations of selected labels and textual descriptions from the SA-OVRS dataset.

## B Details of Training

### B.1 Hyperparameter Setting

### B.2 Templates of Text Instructions

```
"Segment the {object_name} in the provided image."
"Produce a segmentation mask for the {object_name}."
"Output the segmentation mask corresponding to the {object_name}."
"Locate the {object_name} and generate a segmentation mask."
"Provide a segmentation mask for the {object_name}."
```

Table 1: Hyperparameter settings for pre-training and fine-tuning. "Sem." denotes semantic segmentation task and "Ref." denotes referring segmentation task.

| Hyperparameter | Pre-training | Fine-tuning |
|---|---|---|
| Learning Rate | 2e-4 | 1e-4 |
| $\beta_1$ | 0.9 | 0.9 |
| $\beta_2$ | 0.95 | 0.99 |
| Weight Decay | 5e-2 | 0 |
| Gradient Clipping | 1.0 | 1.0 |
| Learning Rate Scheduler | – | WarmupCosineDecay (1% warmup) |
| Precision | `bf16` | `bf16` |
| Epochs | 4 | 10 for Sem. / 20 for Ref. |

Table 2: Model configurations used for autoregressive (AR) training.

| Model | Parameters | Resolution | Global Batch Size |
|---|---|---|---|
| LLaMA Transformer | 0.77B | 256 | 128 |
| | 1.5B | 256 | 128 |
| | 1.5B | 384 | 64 |
| Janus Pro | 1B | 384 | 128 |

```
"Create a mask that highlights the {object_name} from the given
image."
"Output a mask for the {object_name} in the provided image."
"Identify the {object_name} and generate a segmentation mask."
"Show the segmentation mask for the {object_name}."
"Extract the {object_name} and output a segmentation mask."
```

## B.3 DATA AUGMENTATION

Unlike natural language tasks, visual tasks typically have a fixed training dataset, which has an upper limit even when the dataset is large. However, autoregressive models have a strong data memory capability. To mitigate this, we need to implement dynamic data augmentation strategies within each epoch of training. The above ten text templates are one such strategy. Additionally, images will undergo random horizontal flipping during training, provided that the text description does not contain "**left**" or "**right**", along with slight color changes (brightness=0.2, contrast=0.2, hue=0.05). Moreover, we have designed a cropping strategy specifically for segmentation data.

The cropping strategy employs adaptive scaling based on the bounding box of target masks to preserve semantic integrity. When the image width exceeds height, the algorithm calculates scaling ratios along the shorter dimension (height): if the mask height is smaller than the target size, the original height is retained; otherwise, proportional resizing ensures mask dimensions meet the minimum target requirement. A randomized scaling factor is selected within the valid range to maintain aspect ratio while guaranteeing the shorter edge never falls below the target size. Following resizing, connected component analysis is performed on masks to extract region-of-interest boundaries. The cropping coordinates are dynamically constrained to ensure the randomly selected crop window fully contains at least one connected component's minimum bounding rectangle. This spatial constraint mechanism, coupled with aspect ratio-preserved scaling, systematically prevents mask truncation during random cropping operations. The methodology effectively balances data diversity enhancement with rigorous preservation of mask completeness, addressing critical requirements for robust multi-modal segmentation model training.

# C  EVALUATION DETAILS

## C.1  VISUALIZATION RESULTS

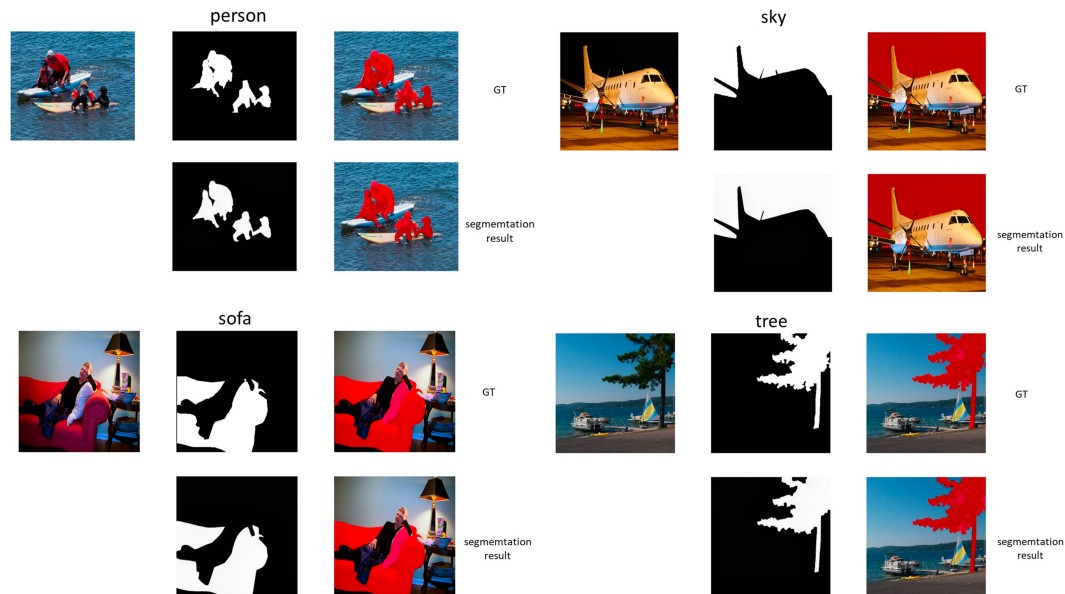

Figure 8: Visualization results on semantic segmentation datasets.

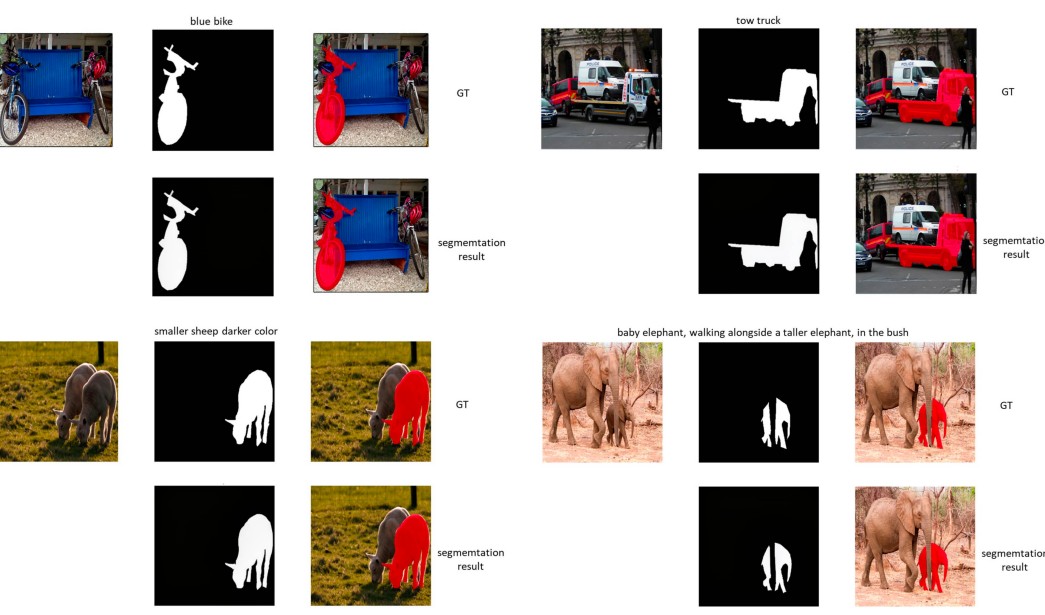

Figure 9: Visualization results on referring segmentation datasets.

## C.2 Failure Cases

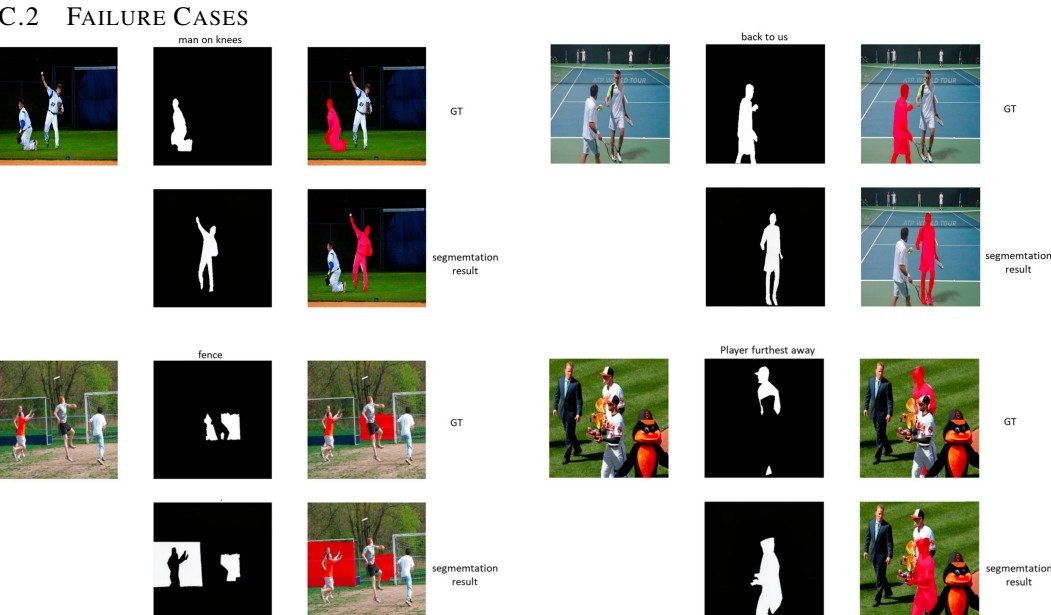

Figure 10: Visualization of failure cases.

## D Further Training of Mask Tokenizer

Table. **??** shows that training the mask tokenizer with mask data can lead to a performance drop. A pre-trained VQGAN, trained on numerous color images, has strong visual feature capturing and pixel reconstruction abilities. Since mask data only has black and white pixels, its features are simpler than color images, making it easier for the pre-trained VQGAN to capture edges. Although training the tokenizer on masked data seems to yield better results, it consumes significant resources. However, the model trained on large amounts of masked data still underperforms the frozen-parameter model. Visual examples are provided in Fig. 11

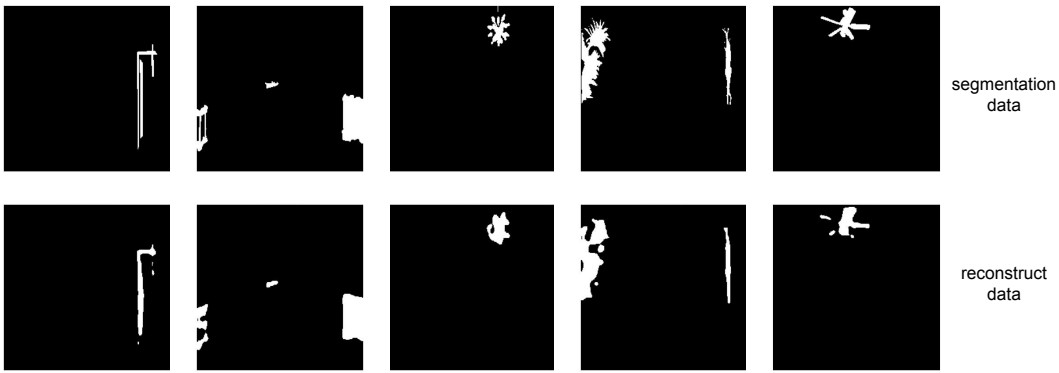

Figure 11: Comparison between the segmentation masks and reconstructed masks. Although the IoU between the reconstructed and original mask is high, many detailed features have been lost.

## E Limitations

Our model can generate complex contours but still needs improvement in understanding intricate language instructions. Also, the fixed resolution hinders the detection of small objects, especially in semantic segmentation datasets with many small regions. In the future, we will explore multi-scale visual generative methods to improve segmentation accuracy for small objects. We plan to

integrate more segmentation tasks and datasets to unify image segmentation within an autoregressive architecture.