# OpenReview forum: "LlamaSeg: Image Segmentation via Autoregressive Mask Generation"
_ICLR.cc/2026/Conference — Submitted to ICLR 2026_

### Official Review · Reviewer_YhD4 · 2025-10-26

**Soundness:** 2
**Presentation:** 3
**Contribution:** 2
**Rating:** 2
**Confidence:** 4

**Summary:**

The paper proposes LlamaSeg, a visual autoregressive model that reformulates image segmentation as visual generation using a LLaMA-style Transformer. Trained on the large-scale SA-OVRS dataset with 2M annotated masks, it supports text-guided segmentation and fine-grained mask generation.

**Strengths:**

1. LlamaSeg introduces the idea of using an image tokenizer to encode segmentation masks, effectively unifying various segmentation tasks within a discrete autoregressive framework.

2. The paper is clearly written and easy to follow.

**Weaknesses:**

1. The comparison baselines are outdated, and LlamaSeg’s segmentation performance is not competitive (e.g., around 56 on RefCOCO), which is significantly lower than recent methods such as Ferret-v2 [1] (≈90).

2. The relatively poor performance raises doubts about whether encoding masks using image tokenizer truly offers advantages over encoding them as discrete position tokens or point sequences. A more detailed comparison and ablation studies (including performance and efficiency) across different mask encoding strategies is needed to justify this design choice.

3. Encoding a single mask requires hundreds of visual tokens, which appears less efficient than directly encoding the mask as a compact sequence of points(represented as position tokens in Kosmos-2).

[1] Ferret-v2: An Improved Baseline for Referring and Grounding with Large Language Models. arXiv preprint arXiv:2404.07973, 2024.

[2] Kosmos-2: Grounding multimodal large language models to the world[J]. arXiv preprint arXiv:2306.14824, 2023.

**Questions:**

1. My main concern lies in the motivation of this work. It remains unclear what specific advantages the use of an image tokenizer offers over existing mask encoding methods. Is it intended to improve performance, efficiency, or both?

---

> ### Author Response · Authors · 2025-11-24
> **Rebuttal by Authors**
>
> Thank you for your detailed review and valuable feedback. We appreciate the opportunity to address your comments.
>
> **W1.The comparison baselines are outdated, and LlamaSeg’s segmentation performance is not competitive**
>
> Our work targets generative, autoregressive pixel-level mask prediction, so the appropriate baselines are other generative segmentation models such as Unified-IO and Unified-IO2. Ferret-v2 is a discriminative grounding model with a completely different objective and architecture, and does not perform autoregressive mask generation. Within the generative setting, LlamaSeg achieves solid results and can further advance this modeling direction.
>
> **W2.Whether image-token mask encoding is truly superior to point- or position-based encodings and requests stronger ablations to justify this choice.**
>
> Although point-based or coordinate sequence encodings are compact, prior work has shown that they are fundamentally unstable in autoregressive generation. Polygon-based models such as PolyFormer [1] report that sequential vertex prediction suffers from ordering ambiguity and strong error propagation, where a single mispredicted point can distort the entire mask boundary.
> In contrast, works that treat masks as image-like discrete token grids show substantially improved stability and reconstruction fidelity. Generative Semantic Segmentation [2] demonstrates that VQ-based visual tokens provide a structurally aligned representation for generative models and enable reliable pixel-level mask reconstruction.
>
> Since our goal is to build a practical autoregressive segmentation component that can be naturally integrated with text and other visual tokens, representational stability is more crucial than sequence compactness. The above findings collectively indicate that image token based mask representations not only offer clear modeling advantages but also align more naturally with the architectural priors of autoregressive next token prediction, making them a more suitable foundation for unified multimodal autoregressive systems.
>
> [1] Xiangtai Li, Jingkang Zhang, Zongxin Yang, Peng Gao, Han Hu, Yuning Jiang, Yue Cao, Jingdong Wang. PolyFormer: Referring Image Segmentation as Sequential Polygon Generation. In: Proceedings of the IEEE/CVF Conference on Computer Vision and Pattern Recognition (CVPR), 2023.
>
> [2] Junsong Wu, Yiyi Zhou, Siming Yan, Junshi Huang, Xuecheng Nie, Hong Zhou, Shuai Yi. Generative Semantic Segmentation. In: Proceedings of the IEEE/CVF Conference on Computer Vision and Pattern Recognition (CVPR), 2023.
>
> **W3.Encoding a single mask requires hundreds of visual tokens, which appears less efficient than directly encoding the mask as a compact sequence of points(represented as position tokens in Kosmos-2).**
>
> Our goal is not to minimize mask length but to integrate dense segmentation into a unified autoregressive generative framework. Although point-based encodings (e.g., position-token sequences in Kosmos-2) are compact, they scale poorly for high-resolution or fine-grained masks: accurate boundaries and thin structures often require hundreds or thousands of points, leading to long and unstable sequences.
>
> In contrast, grid-structured visual tokens provide a more robust representation:
>
> 1.Natural alignment with next-token prediction.
>  The grid structure aligns seamlessly with the autoregressive decoding process used in large language models, avoiding the ambiguity and positional instability inherent in point sequences.
>
> 2.Higher boundary fidelity and spatial consistency.
>  Our experiments show that VQ-based visual tokens lead to sharper boundaries and more reliable mask structures.
>
> 3.Better architectural compatibility.
>  Unlike point sequences, which require additional positional rules or specialized decoding designs, discrete visual tokens can be generated using the same next-token mechanism as text. This makes the approach naturally scalable and easier to integrate into unified multimodal autoregressive models.
>
> Thus, even with more tokens per mask, image-token encoding is more stable, expressive, and architecturally compatible for unified autoregressive modeling.
>
> **Q1.What concrete advantage the image tokenizer has over other mask encodings—performance, efficiency, or something else?**
>
> Our goal is not to build a new segmentation model, but to enable LLMs to generate pixel-level masks within a unified autoregressive framework. Prior encodings such as point sequences or text descriptions are poorly suited for this setting because they fail to produce stable, high-fidelity masks under next-token prediction.
>
> Image-token encoding avoids these issues by providing a dense, expressive, and autoregressively compatible representation that handles complex shapes and fine boundaries. This design prioritizes representational reliability and architectural compatibility, forming a unified interface that naturally extends to tasks like visual editing, spatial reasoning, and image-conditioned generation.

---

> ### Comment · Reviewer_YhD4 · 2025-11-25
>
> Thanks for the rebuttal. The authors provided additional evidence, such as PolyFormer’s theoretical arguments about the shortcomings of point-sequence representations and the advantages of a mask tokenizer. However, given the extremely weak performance of the method(also confirmed by other reviewers), I find this justification unconvincing.
>
> The claim that “prior encodings such as point sequences or text descriptions are poorly suited for this setting because they fail to produce stable, high-fidelity masks under next-token prediction” lacks sufficient justification and is not supported by empirical evidence.
>
> I strongly recommend that the authors **provide experimental results to support their claim**, such as compare different mask-encoding strategies (shown in Figure 1 in the paper) under the same experimental settings, rather than relying solely on theoretical arguments to justify the necessity of a mask tokenizer.

---

### Official Review · Reviewer_v1ac · 2025-10-30

**Soundness:** 3
**Presentation:** 4
**Contribution:** 3
**Rating:** 6
**Confidence:** 2

**Summary:**

The paper present the LlamaSeg network which is a visual autoregressive model that can apply image segmentation tasks with natural language instructions. The proposed model LlamaSeg encode the input as visual tokens and use LLAMA style network for next token prediction.

The paper also introduce a new data annotation framework which contains 2M segmentation masks over 5800 labels (SA-OVRS dataset). The new dataset allow the model to localized object based on text prompts. They also introduce a new metric called, Hausdorff Distance to measure the mask contour fidelity. The evaluation of the proposed model shows that in outperform existing methods.

**Strengths:**

1. The proposed method have unified formulation for multiple segmentation tasks such as - semantic, referring, open-vocabulary in one autoregressive model.

2. The proposed method has strong boundary fidelity which cause due to the use of mask-tokenizer and autoregressive decoding.

3. The new dataset SA-OVRS is a large one, with open-vocabulary supervision which improve the performance in multiple tasks.

**Weaknesses:**

1. The proposed method has lower performance on some tasks when comparing to discriminative models

2. The tokens that used has fixed downsample of ×16, which can miss fine details

3. The usage of autoregressive model has some latency issues which is much slower than discriminative models

**Questions:**

1. What is the latency of the proposed model? Is there any trade of between the performance and the runtime latency?

2. Does using finer stride can improve the results of the proposed mode? for example if you use ×8 in the mask tokenizer does the performance imporve?

3. How does the model behave for out of distribution data such as medical or aerial imagery without finetune?

---

> ### Author Response · Authors · 2025-11-24
> **Rebuttal by Authors**
>
> Thank you for your detailed review and valuable feedback. We appreciate the opportunity to address your comments.
>
> **W1.The proposed method has lower performance on some tasks when compared to discriminative models**
>
> We acknowledge that autoregressive generative methods may not yet match the per-task peak performance of highly optimized discriminative segmentation systems, which are specifically designed for efficiency and accuracy in isolated segmentation scenarios. However, we aim to establish a unified autoregressive framework that enables pixel-level segmentation to be modeled in the same token-based generative paradigm as text, vision, and other structured outputs. This unified formulation is crucial for building general-purpose multimodal models that perform segmentation, captioning, reasoning, region-level prediction, and image generation within a single consistent architecture.
> Within this generative setting, our method delivers substantial improvements over existing generative segmentation approaches, demonstrating that high-fidelity mask prediction can be achieved without bespoke decoders, task-specific architectural branches, or discriminative inductive biases. Importantly, our results show that pixel-level segmentation can be effectively expressed and generated through image tokens, paving the way for future unified multimodal autoregressive models that integrate dense prediction with language and vision generation in a single modeling space.
>
> **W2.The tokens that used has fixed downsample of ×16, which can miss fine details**
>
> We agree that a fixed 16× downsampling may limit the ability to capture fine details. Our primary goal is to demonstrate the feasibility of pixel-level segmentation within an autoregressive framework, and we therefore adopt the standard 16× VQ tokenizer to ensure stable training and inference. Despite this limitation, our experiments already show clear improvements in boundary fidelity, such as in the dAHD metric. We plan to explore higher-resolution tokenizers in future work to further enhance fine-grained detail modeling.
>
> **W3 & Q1.What is the model’s latency, and is there a performance–latency trade-off?**
>
> Autoregressive models are naturally slower than discriminative segmentation models, since they generate outputs token by token. Our goal is not to optimize for maximum inference speed, but to demonstrate that pixel-level segmentation can be unified within an autoregressive framework. The latency of the model therefore follows the expected behavior of AR decoding, and higher resolutions lead to longer sequences and a corresponding performance–latency trade-off. We will include latency statistics and a brief discussion of this trade-off in the final version.
>
> **Q2.Does using a finer stride can improve the results of the proposed mode? For example, if you use ×8 in the mask tokenizer does the performance improve?**
>
> A finer stride may in principle provide more detailed spatial representation, but it would also substantially increase the number of visual tokens and raise the cost of autoregressive generation. In our experiments, increasing the input resolution already leads to consistent improvements, suggesting that finer spatial representation can indeed be beneficial. However, reducing the tokenizer stride from 16 to 8 would greatly lengthen the output sequence and may not yield a proportional gain. We will further investigate the effect of finer strides in future work.
>
> **Q3.How does the model behave for out-of-distribution data, such as medical or aerial imagery without finetune?**
>
> Since our model is trained primarily on natural-image datasets, its zero-shot performance on domains with substantial distribution shift, such as medical or aerial imagery, may degrade without fine-tuning. This behavior is common among vision models trained on natural images. However, the autoregressive formulation and discrete visual token representation used in LlamaSeg are generally transferable, and the model can be adapted to new domains with light-weight fine-tuning. We will include a discussion on out-of-distribution scenarios in the final version.

---

> > ### Comment · Reviewer_v1ac · 2025-11-26
> >
> > Thank you for addressing my concerns and for your valuable answer.

---

### Official Review · Reviewer_A2WG · 2025-10-30

**Soundness:** 3
**Presentation:** 3
**Contribution:** 3
**Rating:** 4
**Confidence:** 3

**Summary:**

This paper introduces LlamaSeg, an autoregressive framework for image segmentation that unifies multiple segmentation tasks under the paradigm of next-token prediction.
The key idea is to reformulate segmentation as visual generation, encoding segmentation masks as discrete visual tokens through a VQGAN tokenizer and generating them autoregressively using a LLaMA-style Transformer.
To support large-scale training, the authors construct SA-OVRS, a new dataset containing 2M segmentation masks annotated with over 5,800 open-vocabulary labels and textual descriptions.
Additionally, a novel evaluation metric, average Hausdorff Distance (dAHD), is proposed to assess contour fidelity of generated masks.
Extensive experiments show that LlamaSeg surpasses existing visual generative models (e.g., Unified-IO and Unified-IO2) on both semantic and referring segmentation benchmarks, while producing finer and more accurate mask boundaries.

**Strengths:**

### 1. Conceptual novelty
Reformulating image segmentation as an autoregressive mask generation problem is a creative and elegant extension of large language model principles to pixel-level prediction.
This perspective bridges the gap between generative modeling and structured visual understanding.

### 2. Unified framework
The proposed approach enables seamless integration of segmentation tasks into LLM-based architectures through a consistent tokenization and generation pipeline.
It provides a promising step toward unifying pixel-level vision tasks with autoregressive multimodal modeling.

### 3. Dataset contribution
The introduced SA-OVRS dataset offers large-scale, open-vocabulary segmentation data paired with rich textual descriptions.
This resource can support future research in open-vocabulary and multimodal segmentation.

### 4. Evaluation rigor
The introduction of the dAHD metric is an insightful addition, providing a more nuanced measure of boundary accuracy and contour fidelity compared to traditional IoU-based metrics.

### 5. Empirical validation
Comprehensive experiments across multiple benchmarks and datasets demonstrate consistent quantitative and qualitative improvements, validating the effectiveness of the proposed framework.

**Weaknesses:**

### 1.  Limited scope and contribution
The contribution feels more foundational within a narrow scope rather than broadly transformative.  The method primarily focuses on segmentation and language alignment, without clear extensions to other modalities or tasks such as vision-language reasoning, instruction following, or general multimodal generation.  Compared with highly integrative multimodal frameworks like Unified-IO, 4M-21 (Bachmann, Roman, et al. "4m-21: An any-to-any vision model for tens of tasks and modalities." Advances in Neural Information Processing Systems 37 (2024): 61872-61911.), this work appears less comprehensive and serves more as an initial step toward unifying segmentation within the LLM paradigm rather than a fundamentally new multimodal foundation.

### 2. Incomplete comparison with recent foundation models
Beyond Unified-IO, there exist broader any-to-any vision models such as 4M-21, which support a wider range of tasks and modalities while achieving comparable semantic segmentation results.  The paper should include a direct comparison and a detailed discussion to better contextualize its contribution relative to such works.

### 3. High complexity and unclear efficiency
The proposed segmentation process is quite complex. It involves label generation, mask matching, and a separate inference step for each mask.  This design is inefficient for dense semantic segmentation scenarios.  The paper should report inference time, throughput, and computational cost to clarify the practical feasibility of the approach.

**Questions:**

1. Could the authors clarify whether the model can generalize beyond segmentation to other vision-language tasks, such as referring expression comprehension or open-ended visual reasoning?

2. The segmentation process appears computationally heavy, requiring separate inferences for each mask.  Could the authors provide concrete runtime statistics (e.g., FPS, latency per image, GPU hours) and discuss possible optimizations for dense segmentation settings?

3. Since the model leverages VQ-based mask tokenization, how sensitive is it to the quality of the VQ tokenizer?  Would retraining or substituting the tokenizer significantly impact segmentation accuracy?

---

> ### Author Response · Authors · 2025-11-24
> **Rebuttal by Authors**
>
> Thank you for your detailed review and valuable feedback. We appreciate the opportunity to address your comments.
>
> **W1:Limited scope and contribution**
>
> Our work focuses on a core challenge that current autoregressive models still struggle with: generating dense visual outputs, such as pixel-level segmentation masks, within a next-token prediction framework. A central contribution of this work is to demonstrate that high-fidelity pixel-level segmentation can indeed be performed in a fully autoregressive, purely token-based manner, which has not been shown before.
> Our contributions are as the following points:
>
> 1.We provide the first practical, end-to-end, purely token-based autoregressive method for segmentation.
>
> 2.We introduce SA-OVRS, a large-scale open-vocabulary dataset that can be used by future AR models.
>
> 3.Our approach achieves clearly better results than existing generative segmentation models on multiple benchmarks.
>
> Moreover, our method is not limited to segmentation. The visual token generation mechanism can naturally extend to spatial reasoning, visual editing, image-conditioned generation, and other dense prediction tasks, making it a useful component for future unified multimodal AR models.
>
> **W2. Incomplete comparison with recent foundation models**
>
> In our experiments, we primarily compared with Unified-IO and Unified-IO2 because, similar to our work, they model segmentation as a generative output, making them directly comparable under the same “visual generation / autoregressive” framework.
> We agree that it is important to better position our work within the broader family of unified multimodal models. In the final version, we will expand the Related Work section to include a more detailed discussion of 4M-21 and other any-to-any models.
>
> **W3 & Q2.High complexity and unclear efficiency**
>
> We believe there is a misunderstanding. The steps mentioned by the reviewer—such as label generation and mask matching—are part of the offline data construction for SA-OVRS only. They are not part of the model and never occur during inference, so they do not affect efficiency. At inference time, the pipeline is straightforward: given an image and text, the model performs a single forward pass to autoregressively generate the full mask token sequence, which is then decoded by VQGAN into the final segmentation map. No extra modules, no SAM, no polygon post-processing, and no multi-stage or per-object steps are involved.
>
> **Q1. Generalization Beyond Segmentation**
>
> Although our model is primarily evaluated on semantic segmentation and referring expression segmentation, its formulation is not limited to these tasks. Representing masks as discrete visual tokens naturally aligns with the unified token interface that underlies recent autoregressive multimodal models such as Chameleon [1] and Janus-Pro [2]. These models demonstrate that once visual information is expressed in a tokenized form, it can be processed and generated in the same autoregressive pipeline as text. This property makes our visual-token formulation inherently extensible to a broader set of vision–language tasks.
> In our experiments, we have already validated generalization to referring expression comprehension, achieving results on RefCOCO that surpass prior generative approaches (Table 2). Because the model jointly parses language and produces grounded visual outputs, it can naturally serve as a modular component for more advanced tasks such as open-ended grounding, spatial reasoning, and visually conditioned generation, following how token-based autoregressive frameworks have been used to support multi-task multimodal learning in prior work.
>
> [1] Chameleon Team. Chameleon: Mixed-Modal Early-Fusion Autoregressive Models. arXiv preprint arXiv:2405.12130, 2024.
>
> [2] Chen, J., Ye, Z., Wang, S., et al. Janus-Pro: Unified Multimodal Understanding and Generation via Autoregressive Modeling. arXiv preprint arXiv:2406.06331, 2025.
>
> **Q3. Sensitivity to VQ-Based Mask Tokenizer**
>
> We have actually conducted relevant ablation studies in the paper (Table 4), and the results show that the model is not particularly sensitive to the quality of the VQ-based mask tokenizer. We experimented with further retraining the original VQGAN tokenizer for 5k and 10k steps, but this led to worse results—lower IoU and higher mAHD. This suggests that the pretrained tokenizer is already stable, and excessive fine-tuning disrupts its codebook structure. These findings indicate that the model’s performance is mainly determined by its autoregressive generation capability rather than by the specific choice or fine-tuning of the tokenizer.

---

> > ### Comment · Reviewer_A2WG · 2025-11-25
> >
> > Thanks for the rebuttal. I share the same opinion as the other reviewer and strongly recommend that the authors provide experimental results demonstrating the efficiency and generalization of the method beyond segmentation.

---

### Meta-Review · Area_Chair_evPz · 2026-01-01

**Summary:**

The paper proposes LlamaSeg, a framework that reformulates image segmentation as a visual autoregressive generation task. By encoding masks as discrete visual tokens (via VQGAN) and utilizing a LLaMA-style transformer, the method aims to unify segmentation within a next-token prediction paradigm. The authors also contribute a large-scale dataset (SA-OVRS) and a boundary-focused metric ($d_{AHD}$).

While the reviewers acknowledged the conceptual novelty of the unified framework and the value of the proposed dataset, the consensus leans towards rejection. The primary concerns revolve around the significant performance gap compared to state-of-the-art discriminative models, the lack of empirical justification for the chosen mask representation over more efficient alternatives, and concerns regarding the inference efficiency of autoregressive modeling for dense prediction. The rebuttal failed to provide the requested empirical ablations and efficiency statistics, relying instead on theoretical arguments that did not satisfy the reviewers. Therefore, the paper is not ready for acceptance in its current form.

**Reviewer Concerns:**

Addressed by Rebuttal:
- Dataset Construction vs. Inference: The authors clarified for Reviewer A2WG that the complex pipeline involving label generation and matching is part of the offline dataset creation, not the inference process.

- Tokenizer Sensitivity: The authors addressed questions from Reviewers A2WG and v1ac regarding the sensitivity of the VQ-tokenizer, providing ablation data (Table 4) to show that the pre-trained tokenizer is stable and that further fine-tuning degrades performance.

Outstanding / Not Adequately Addressed:

- Comparison with SOTA & Baselines: Reviewer YhD4 strongly criticized the "extremely weak" performance (e.g., \~56 on RefCOCO) compared to recent baselines like Ferret-v2 (\~90). The authors' counter-argument—that they should only be compared to other generative models like Unified-IO—was viewed as unconvincing given the magnitude of the performance gap.

- Justification of Mask Representation (Ablation): Reviewer YhD4 requested an empirical ablation comparing the proposed image-token encoding against point/coordinate-based encodings (used in PolyFormer/Kosmos-2) to justify the efficiency/performance trade-off. The authors provided theoretical arguments and citations rather than the requested experimental evidence, which the reviewer explicitly found insufficient.

- Efficiency and Latency: Reviewers A2WG and v1ac raised concerns about the computational cost of generating hundreds of visual tokens for a single mask. While the authors acknowledged the trade-off and promised statistics in the "final version," they did not provide the concrete runtime/FPS data requested during the discussion phase to prove practical feasibility.

- Generalization Scope: Reviewer A2WG questioned the "unified" claim, noting that the model is currently limited to segmentation. The reviewer recommended demonstrating generalization to other vision-language tasks (e.g., visual reasoning) to justify the complex generative architecture, which remains outstanding.

**Reviewer Scores:**

- Reviewer A2WG: 4 (Marginally Below Acceptance). While this reviewer appreciated the clarifications on the pipeline, their final comment strongly reiterated the need for efficiency statistics and generalization results. Since these were not provided in the rebuttal, their score would likely remain unchanged.

- Reviewer v1ac: 6 (Marginally Above Acceptance). This reviewer was the most positive and thanked the authors for the responses. However, they acknowledged the latency and resolution limitations. Without the strong support of other reviewers regarding performance, they would likely maintain their score or potentially drop to a 5, but the AC would estimate they hold at 6 based on their polite closing comment.

- Reviewer YhD4: 2 (Reject). This reviewer explicitly stated in their post-rebuttal comment that the justification for the mask encoding was unconvincing and the performance was too weak. They strongly recommended experimental results over theoretical arguments. Their score would definitely remain a strong reject.

---

### Decision · Program_Chairs · 2026-01-26

Reject